# Perceptual Group Tokenizer:
# Building Perception with Iterative Grouping

**Zhiwei Deng, Ting Chen, and Yang Li**
Google Research and Deepmind

## Abstract

Human visual recognition system shows astonishing capability of compressing visual information into a set of tokens containing rich representations without label supervision. One critical driving principle behind it is perceptual grouping (Palmer, 2002; Wagemans et al., 2012; Herzog, 2018). Despite being widely used in computer vision in the early 2010s, it remains a mystery whether perceptual grouping can be leveraged to derive a neural visual recognition backbone that generates as powerful representations. In this paper, we propose *the Perceptual Group Tokenizer*, a model that entirely relies on grouping operations to extract visual features and perform self-supervised representation learning, where a series of grouping operations are used to iteratively hypothesize the context for pixels or superpixels to refine feature representations. We show that the proposed model can achieve competitive performance compared to state-of-the-art vision architectures, and inherits desirable properties including *adaptive computation without re-training*, and interpretability. Specifically, Perceptual Group Tokenizer achieves 80.3% on ImageNet-1K *self-supervised learning* benchmark with linear probe evaluation, establishing a new milestone for this paradigm.

## 1 Introduction

Visual recognition mechanisms matter. The pursuit of advanced vision algorithms that encode an image to meaningful representations dates back to late 80s, with two paradigms marking the progress over the past 40 years: feature detection (LeCun et al., 1998; Lowe, 2004; He et al., 2016; Liu et al., 2022b) and perceptual grouping (Shi & Malik, 2000; Uijlings et al., 2013; Arbeláez et al., 2014), where feature detection focuses on specific distinctive patterns, while perceptual grouping considers similarities among all pixels to produce a compact set of tokens as proxies for image representation. Ever since the surge of deep learning, feature detection has predominated the vision field and become the main principle behind representation learning backbone designs and made impressive progress (Simonyan & Zisserman, 2014; Szegedy et al., 2015; He et al., 2016; Chen et al., 2017; Tan & Le, 2019; Qi et al., 2020; Liu et al., 2022b). The success of the former paradigm is, although striking, raising the question of whether perceptual grouping can also be used as the driving principle to construct a visual recognition model.

Different from detecting and selecting distinctive features, perceptual grouping emphasizes on learning feature space where similarity of all pixels can be effectively measured (Uijlings et al., 2013; Arbeláez et al., 2014). With such a feature space, semantically meaningful objects and regions can be easily discovered with a simple grouping algorithm and used as a compact set to represent an image (Uijlings et al., 2013; Arbeláez et al., 2014; Locatello et al., 2020). This indicates that image understanding is essentially "pixel space tokenization", and being able to produce generalizable feature representations is tightly connected to whether the correct contextual pixels are binded together (Hinton, 2022; Culp et al., 2022).

The intriguing properties of perceptual grouping, including natural object discovery, deep connections with information theory and compression (Ma et al., 2007), and association with biological vision system (Herzog, 2018) or cognitive science explanations (Palmer, 2002), have led to a strong revival recently under deep learning frameworks (Locatello et al., 2020; Elsayed et al., 2022; Xu et al., 2022; Wu et al., 2022; Biza et al., 2023). However, these methods are either still focusing on small or toy datasets (Locatello et al., 2020; Chang et al., 2022; Biza et al., 2023), or used as an auxilliary component (Xu et al., 2022; Ke & Yu, 2022; Seitzer et al., 2022) to strengthen exist-

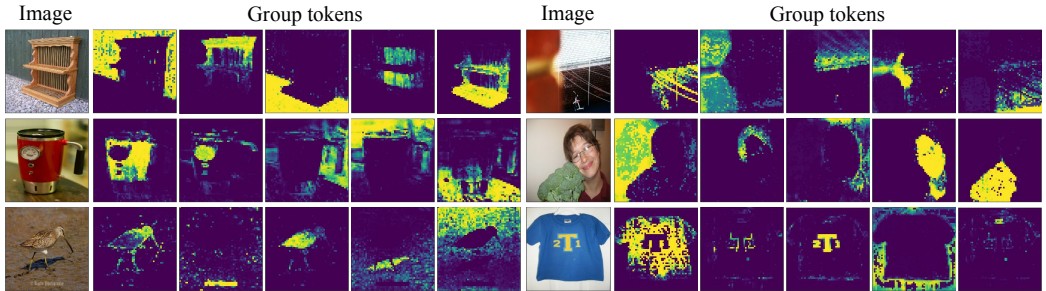

Figure 1: **Perceptual Group Tokenizer** is entirely driven by **grouping operations** to perform representation learning. Group tokens (discovered objects) are shown above. See more in the appendix.

ing vision architectures for increased interpretability. Whether perceptual grouping can be used to build models and learn representations that are as informative and expressive as those learned by state-of-the-art vision architectures remains an open question.

In this paper, we propose *Perceptual Group Tokenizer*, a model trained under a *self-supervised learning* framework, which builds visual representation *entirely based on perceptual grouping operations*. Given an image, the core of our model is to understand each pixel or patch through hypothesizing its contexts with grouping operations. Starting from given input patches, the grouping operation performs an iterative binding process onto a set of randomly sampled group tokens to determine the affinity groups based on similarities. The group tokens are then used as hypothesized contexts to refine the feature representation for the image. We show that applying this simple principle can already produce expressive representations and works well with self-supervised pretraining on a large vision dataset.

The grouping operation is also closely related to self-attention, a highly popular method commonly used in modern vision backbones. We build connection between the proposed grouping operation and self-attention and show that, if group tokens are treated as communication channels, self attention can potentially automatically emerge during learning processes as a special case, while the grouping operation can produce even richer interactions among tokens. Under this viewpoint, ViT (Dosovitskiy et al., 2020) can be considered as a grouping backbone, with a fixed number of grouping slots equal to the number of input tokens, and the binding is achieved through stacking more than one layer with non-shared weights. This provides one explanation on why grouping mechanism can be effective on visual representation learning and has the potential to be a promising competitive paradigm for vision architecture designs.

The primary contribution of this work is proposing a new architecture derived purely by perceptual grouping that achieves competitive performance compared to other state-of-the-art architectures on *self-supervised learning* benchmarks, contributing to a new paradigm of developing vision architectures. The model has several key differences and advantages over ViT, including (1) explicit separating out the "group token" concept to allow for automatic image parsing and flexible customization on the number of groups without being binded to the number of patches; (2) much less peak memory usage during inference time given the same number of input tokens; (3) adaptive computation without re-training the model, leading to flexible usage according to domains and computes.

## 2   RELATED WORKS

**Vision architectures.** There are two main frameworks for vision backbones. The first framework is Convolutioinal neural networks, which rely on local filters, sliding windows and translational equivariance to perform representation learning. Since the introduction of ConvNets in 1980s, ConvNet was repopularized by AlexNet (Krizhevsky et al., 2012). The line of ConvNet is a classical inheritance from traditional feature detection methods (Lowe, 2004; Dalal & Triggs, 2005; Rosten et al., 2008), where instead of hand crafting features, an overcomplete set of filters are automatically learned to obtain high-response regions. The object understanding is built along the depth axis (Simonyan & Zisserman, 2014; Szegedy et al., 2015; He et al., 2016), with early layers capturing low-level parts and higher-level layers producing object structure representations (Zeiler & Fergus, 2014; Zhou et al., 2014; Yosinski et al., 2015; Bau et al., 2017). In the feature detection framework,

not every pixel is worth being used depending on particular tasks, leading to difficulty in obtaining representation for each pixel.

Recently, Vision Transformer (ViT) (Dosovitskiy et al., 2020), a second vision backbone framework, shows impressive performance and has surpassed ConvNet on visual recognition. The core of ViT is the iterative applying of self-attention operations (Vaswani et al., 2017; Dosovitskiy et al., 2020). A direct usage of ViT on small patches (thus a high-resolution grid) is extremely computationally expensive due to its associated quadratic cost. Therefore, a common practice is often partitioning the image into large non-overlapping patches (Dosovitskiy et al., 2020; Touvron et al., 2021), or constrain the operation to local regions (Liu et al., 2021).

**Self-supervised learning.** The field of representation learning has seen significant interest in self-supervised learning during the past few years. The main evaluation results using linear probe on ImageNet benchmarks is approaching the results obtained by supervised learning (Oquab et al., 2023). Contrastive representation learning is the early method that shows promising results (Oord et al., 2018; Chen et al., 2020a; Tian et al., 2020). BYOL (Grill et al., 2020) and DINO (Caron et al., 2021) propose to use a moving average target of an online network to perform self representation matching. Masked image modeling also shows to be effective on representation learning, where the masking is either at the pixel level (He et al., 2022) or the learned codebook level (Bao et al., 2021).

**Object discovery.** The perceptual grouping is essentially performing "object and stuff" discovery in the pixel space. It has broad connections with the early works in computer vision (Shi & Malik, 2000; Uijlings et al., 2013; Levinshtein et al., 2013; Arbeláez et al., 2014; Pont-Tuset et al., 2016), the recent progress on object-centric representation (Burgess et al., 2019; Locatello et al., 2020; Chang et al., 2022; Hinton, 2022; Hénaff et al., 2022; Culp et al., 2022; Elsayed et al., 2022), and biological or neural mechanisms on perceptual grouping (Palmer, 2002; Wagemans et al., 2012; Herzog, 2018; Kim et al., 2019). Despite the early popularity of perceptual grouping methods on various computer vision tasks (Shi & Malik, 2000; Uijlings et al., 2013; Levinshtein et al., 2013; Krähenbühl & Koltun, 2011), it has not attracted significant attention until several recent works that apply it as a side component on top of another main backbone (Seitzer et al., 2022; Liu et al., 2022a; Xu et al., 2022; Ke & Yu, 2022). Some relevant works demonstrate alternative possibilities in architecture design, but only uses cross attention without refining the patch feature space (Jaegle et al., 2021), or apply it on diffusion tasks (Jabri et al., 2022). Other methods also attempt to use ad-hoc sparsification methods on top of ViT (Rao et al., 2021; Yin et al., 2022; Bolya et al., 2023) for efficiency and are orthogonal to our work. A most related work (Ma et al., 2023) focuses on supervised learning and relies on fixed-center pooling and less standard operations. In our proposed model, we adopt a design as ViT except for self attention, and highlight several key technical contributions, including multi-grouping with multi-seeding, adaptive computation without re-training, and other design choices for self-supervised representation learning.

## 3 MODELS

In this section, we introduce Perceptual Group Tokenizer (PGT), a visual recognition architecture entirely driven by perceptual grouping principles. We discuss the core operations for grouping in section 3.1, the building blocks and network architectures in section 3.2, the loss function used for self-supervised learning in section 3.3, and the connections with other models in section 3.4.

### 3.1 PERCEPTUAL GROUPING

We start with introducing notations for our method. Given an image $x \in \mathbb{R}^{H \times W \times C}$, we first reshape it as a sequence of small patches[1]. Each patch $x_p \in \mathbb{R}^{h \times w \times c}$ has spatial shape $h \times w$, where $h \ll H$ and $w \ll W$, leading to $N = \frac{HW}{hw}$ number of patches per image. To represent a patch, we embed it into a high-dimensional vector $h \in \mathbb{R}^d$. The set of embedded tokens $\{h_i\}^N$ is referred to as *input tokens* in later parts, and used as inputs for the following grouping blocks.

**Feature refinement through hypothesizing contexts.** Individual pixels do not have meanings without putting it into contexts. At a high level, image understanding or feature learning is equivalent to binding the correct contextual pixels at all locations. The core idea of our model is to generate many (e.g. over-complete w.r.t number of objects in the image) hypothesized contexts and use the

---

[1]We use $4 \times 4$ patches as inputs in this work. Note that our method is generalizable to either pure pixels or other forms of superpixels given a proper patch-to-vector embedding layer.

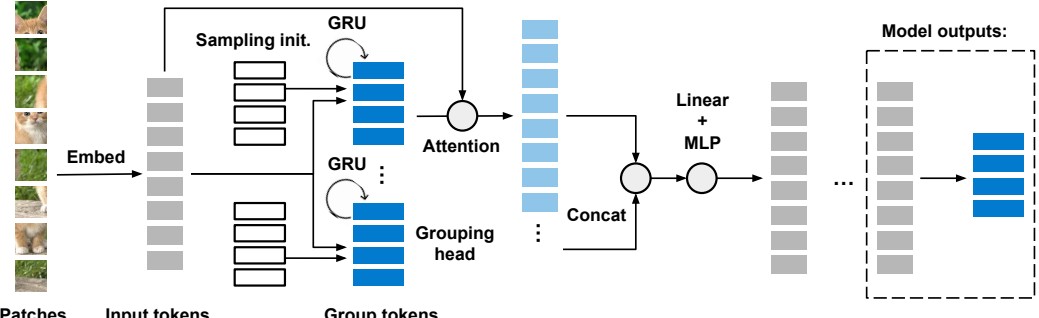

Figure 2: Perceptual Group Tokenizer takes in a sequence of patches (or pixels), generates high-dimensional embedding vectors for all patches, then them passes through a series of grouping layers to refine the embedding vectors as feature representations. Each grouping layer performs $K$ rounds of binding from input tokens to group tokens. To consider various grouping possibilities, multiple grouping heads are adopted. Each group token provides a useful context for input tokens for feature refinement. The final output of the model contains refined input token, group tokens, and assignments between input tokens and groups tokens.

hypothesized contexts as cues to refine the feature representation of each patch. This process is achieved through a grouping module. Given input tokens $\{h_i\}^N$, the grouping module starts from a set of random samples (referred as *group tokens*) from a random distribution, then performs binding process to aggregate information from input tokens to the group tokens, and ends up with a set of group tokens $c^* = \{c_j^*\}_{j=1}^M$ representing hypothesized contexts among input tokens. The relation between $h_i$ and $c_j$ is soft assigment, indicating how likely an input token belongs to that context. Note that there are often *various ways of generating groupings for an image*, e.g. different semantics, colors, textures, etc., we propose the "multi-grouping operation" to hypothesize rich contexts for tokens. The overall model is shown in figure 2.

**Multi-grouping operation.** The building block of our model is the multi-grouping operation $\mathcal{G}$, which contains multiple heads to perform the binding process in parallel. This design encourages the model to consider multiple ways of generating groups under different projection spaces. Each head owns a separate Gaussian distribution with learnable means and variance, similar to (Kingma & Welling, 2013; Locatello et al., 2020). Starting from a set of randomly sampled initial group tokens $c_{\mathrm{HEAD}}^{(0)} \sim p_{\mathrm{INIT}}(\cdot)$, the grouping operation uses doubly normalized attention weights to aggregate information from $h$, and the produced group tokens $c_{\mathrm{HEAD}}^{(1)}$ are used for the next round binding. The attention normalization and feature projection are performed in all heads separately.

$$c_{\mathrm{HEAD}}^{(1)} \quad = \quad \mathcal{G}(c_{\mathrm{HEAD}}^{(0)}, h; \theta) \tag{1}$$
$$\ldots$$
$$c_{\mathrm{HEAD}}^* = c_{\mathrm{HEAD}}^{(K)} \quad = \quad \mathcal{G}(c_{\mathrm{HEAD}}^{(K-1)}, h; \theta) \tag{2}$$

where after K steps the final group tokens $c^* = c^{(K)}$ is obtained, and $\theta$ is learnable parameters in $\mathcal{G}$. The grouping operator is summarized in algorithm 1.

The sampling distribution $p_{\mathrm{INIT}}(\cdot)$ for initializing group tokens $c_{\mathrm{HEAD}}^{(0)}$ needs to be lightweight. We explore two variations: (1) Gaussian distribution $p(\mu_{\mathrm{HEAD}}, \sigma_{\mathrm{HEAD}})$ with learnable means and variance, and a one-step normalizing flow module that transforms a unit Gaussian noise to a sample that follows more complex distributions. More details can be found in the appendix in section A.1

**Implicit differentiation.** The iterative grouping process unrolls $K$ steps per operation and leads to heavy burden in the training computation graph. Instead of explicitly backpropagating through the unrolled graph, we follow (Chang et al., 2022) and treat the multi-grouping process as a fixed point iteration per head. The gradient in the backpropagation is approximated using first-order Neumann series, which can be simply achieved by detaching the output before the final iteration.

### 3.2 NETWORK ARCHITECTURE

Similar to standard ViT, our model refines the hidden representation $h$ using $L$ model layers. We use $h^l$ to denote the representation after each layer, and explain the design in this section.

---

**Algorithm 1** Multi-grouping operation using $\mathcal{G}$.

---

```
def multi_grouping(h_key, h_value, steps, num_tokens, num_heads):
    """ Input tensors:
        h_key and h_value are projected multi-head tensors with shape [num_heads x N x d].
    """
    # Initial M group tokens.
    group_tokens = sampling_distribution(nsamples=num_tokens, choice='Gaussian') # or 'Flow'
    group_tokens = group_tokens.reshape(num_heads, num_tokens, d) #[num_heads x M x d]

    # Binding process.
    for step in range(steps):
        # Implicit differentiation.
        if step == steps - 1:
            group_tokens = stop_gradient(group_tokens)
        """ The following is a one-step grouping operation. """
        # Attention operation for group assignment.
        attn_matrix = attention(group_tokens, h_key) #[num_heads x N x M]
        attn_matrix /= attn_matrix.sum(-2, keep_dim=True)
        h_updates = einsum("hij,hid->hjd", attn_matrix, h_value) #[num_heads x M x d]
        group_tokens = gru_cell(h_updates, group_tokens)
        # Grouped mlp/layernorm performs independent mlp/layernorm for each head.
        group_tokens = grouped_mlp(grouped_layer_norm(group_tokens)) + group_tokens

    return group_tokens
```

---

**Grouping layer.** Each grouping layer takes in $h^{l-1}$ as input, and uses the grouping operation in equation 1 to generate group tokens $c^*_{\text{HEAD}} = \{c^*_{j,\text{HEAD}}\}^M_{j=1}$. To use the group tokens to provide context for each $h^{l-1}_i$, we perform another attention operation to obtain the attention matrix (only normalized over group token axis) $A \in \mathbb{R}^{N \times M}$ representing the assignment from input tokens to group tokens, and aggregate the feature back to the input token space:

$$h^l_{\text{HEAD}} = A[c^*_{1,\text{HEAD}}; c^*_{2,\text{HEAD}}; ...; c^*_{M,\text{HEAD}}] \qquad (3)$$

$$h^l = \text{Linear}([h^l_{\text{HEAD}_1}; ...h^l_{\text{HEAD}_H}]) \qquad (4)$$

$$h^l = h^{l-1} + \text{MLP}(\text{LN}(h^l)) \qquad (5)$$

This layer definition follows the standard ViT layer as close as possible, where features from each head are aggregated through concatenation and a linear layer transformation. Each token $h$ is further refined using a follow up multi-layer perceptron.

**Grouping blocks.** Similar to previous architecture designs (He et al., 2016; Liu et al., 2021). we define blocks for the model. One block contains multiple grouping layers that share the same hyperparameters setups, i.e. the number of group tokens, and group token dimensions. The full model contains three grouping blocks. This increases the flexibility when exploring model design spaces.

### 3.3 SELF-SUPERVISION LOSS

We strictly follow the student-teacher self-supervision loss (Caron et al., 2021; Oquab et al., 2023), and use a moving average of online network (student model) as the teacher model to perform representation learning. To summarize group tokens outputed from the final layer, we use one multi-head attention layer with a learnable token to attend to all group tokens. The produced single vector is treated as the feature representation for the image and is input to the loss function.

### 3.4 DISCUSSION

Our proposed model, perceptual group tokenizer, does not contain self-attention operations and purely relies on grouping operations. In this section, we link the grouping process to several techniques and discuss the rationale on why this model can be effective on representation learning.

**Group tokens as "communication channels".** The core of feature representation learning is how information is exchanged among pixels. In perceptual grouping backbones, we can consider the set of group tokens as communication channels, where information from different input tokens are aggregated in various ways. Each group token represents a high-order channel that links input tokens with high affinity under certain projected space to exchange information among them. As a thought experiment, if each input token is

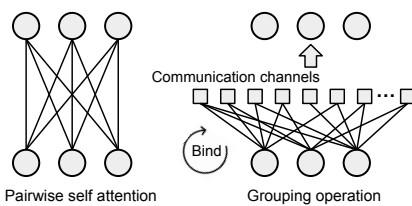

Figure 3: Operation comparison.

---

solely assigned to a different group token (given enough group tokens), then the perceptual grouping layer is equivalent to one self attention layer (up to some engineering design difference). While self attention layers mainly rely on pairwise communications, grouping operation, hypothetically, can automatically learn and emerge both pairwise and higher-order information exchange through the group token communication channels. This can also be linked to traditional *factor graphs* in probabilistic graphical models. Through the lens of that, grouping is forming factor nodes automatically through the learning processes. With a properly designed loss and grouping operation, it has the potential to be more effective if adopting a per-layer comparison with self-attention operations.

**Efficiency.** Due to the flexibility in customizing number of group tokens (controlled by initial number of samples), grouping operation does not require a strict $O(N^2)$ operation and is $O(NM)$ on complexity. Furthermore, we show that *in inference time*, number of group tokens can *even be adaptively customized*, given an already trained model.

## 4 EXPERIMENTS

We evaluate the representation learned by our model on standard benchmarks based on the ImageNet-1K dataset. We also explore and analyze the design space of perceptual group tokenizer in section 4.2, investigate its adaptive computation ability in section 4.3, demonstrate its generalization ability on semantic segmentation in section 4.4, and visualize learned attentions in section 4.5.

### 4.1 MAIN RESULTS

**Setup.** The widely-adopted standard benchmark for evaluating self-supervised learning methods is ImageNet ILSVRC-2012 (ImageNet-1K) (Russakovsky et al., 2015). Performance of models are measured by top-1 classification accuracy. The pre-trained backbones are frozen, with a linear classifier trained on top. For fair comparison, we follow the standard data augmentation used in (Caron et al., 2021), with the same number of global views and local views. The model is optimized using AdamW (Loshchilov & Hutter, 2018) with learning rate 0.0005 and 1024 batch size for 600 epochs, trained with TPUv5 for 21k core hrs (512 cores for 41 hrs). We use 4×4 patches as image tokens, which keeps as much details as possible while maintaining reasonable computation costs.

| Method | Arch | Param. | Linear probe (top-1 acc) |
|---|---|---|---|
| *(Other backbones with different losses within the same batch of DINO for reference)* | | | |
| SCLR (Chen et al., 2020a) | RN50W4 | 375 | 76.8 |
| SwAV (Caron et al., 2020) | RN50W2 | 93 | 77.3 |
| BYOL (Caron et al., 2020) | RN50W2 | 93 | 77.4 |
| SwAV (Caron et al., 2020) | RN50W5 | 586 | 78.5 |
| BYOL (Caron et al., 2020) | RN50W4 | 375 | 78.6 |
| iBOT (Zhou et al., 2021) | ViT-B/16 | 85 | 79.5 |
| BYOL (Caron et al., 2020) | RN200W2 | 250 | 79.6 |
| SCLRv2 (Chen et al., 2020b) | RN152w3+SK | 794 | 79.8 |
| BEiTv2 (Peng et al., 2022) | ViT-B/16 | 85 | 80.1 |
| *(Fair comparison under the DINO loss and framework)* | | | |
| DINO (Caron et al., 2021) | ViT-S/8 | 21 | 79.7 |
| Ours (PGT$_G$-S-1024) | PGT-S | 34 | 79.8 |
| DINO (Caron et al., 2021) | ViT-B/16 | 85 | 78.2 |
| DINO (Caron et al., 2021) | ViT-B/8 | 85 | 80.1 |
| Ours (PGT$_G$-B-256) | PGT-B | 70 | 79.7 |
| Ours (PGT$_G$-B-512) | PGT-B | 70 | 79.9 |
| Ours (PGT$_G$-B-1024) | PGT-B | 70 | 80.1 |
| Ours (PGT$_F$-B-256) | PGT-B | 115 | 80.0 |
| Ours (PGT$_F$-B-512) | PGT-B | 115 | 80.1 |
| Ours (PGT$_F$-B-1024) | PGT-B | 115 | **80.3** |

Table 1: Comparison with strong baselines on ImageNet-1K under linear probe evaluation protocol. PGT$_{\text{DIST}}$-B-$X$ represents $X$ number of group tokens per grouping layer in inference (same trained model with 256 tokens is used). DIST: the distribution choice for group token initialization. G and F represent Gaussian and Flow, respectively. Our model achieves 80.3%, competitive with state-of-the-art vision backbones.

|  | Descend | Flat | Ascend |
|---|---|---|---|
| Token size | $\begin{bmatrix} 576, 384, 192 \end{bmatrix}$ | $\begin{bmatrix} 384, 384, 384 \end{bmatrix}$ | $\begin{bmatrix} 192, 384, 576 \end{bmatrix}$ |
| Accuracy | 62.0 | 63.1 | **63.4** |
| Token shape | $\begin{bmatrix} 192, 128, 64 \end{bmatrix}$ | $\begin{bmatrix} 128, 128, 128 \end{bmatrix}$ | $\begin{bmatrix} 64, 128, 192 \end{bmatrix}$ |
| Accuracy | 63.6 | **63.7** | 63.1 |

Table 2: Exploring the design choices for PGT. Token size: dimensions for group tokens in three grouping blocks. Token shape: number of tokens for group tokens in three grouping blocks. Accuracy measured on ImageNet-1K under linear probe protocal. Results indicate progressively large group token dimensions with flat or descend number of tokens arrangements work the best.

**Architecture details.** In the experiments, we mainly evaluate two variants of PGT: the main model and a tiny version for exploring design choices. On the ImageNet-1K benchmark, we report the performance metrics of our main model. Three grouping blocks are used, with 10 grouping layers in each block. The dimension for input token is 384, with 256 group tokens per layer. The dimensions for group tokens are 98, 192, and 288 for the three blocks, respectively. There are 6 grouping heads used. For number of grouping iterations, we observe three rounds are sufficient to achieve good performance. The MLP hidden size for each layer is 384 as well, i.e., the MLP multiplication factor is 1. The final multihead attention layer uses a learnable token with 2048 dimensions to summarize all group tokens outputs from the model.

The main results are summarized in table 1. We mainly compare with ResNet and ViT backbones, the two main stream vision architectures to show that perceptual grouping architecture can also achieve competitive results on the challenging ImageNet-1K benchmark. Although our model is trained with 256 group tokens, the model can use different numbers of group tokens in inference (more experiments in section 4.2). We evalaute PGT with 256, 512, and 1024 number of group tokens and observe that the model can achieve 80.3% top-1 accuracy, showing the self-supervised learned feature of PGT is as good as the ones learned by ViT architectures.

## 4.2 ABLATIONS

To explore design choices of PGT, we use a tiny version of PGT with 3 blocks, 2 layer in each block (6 layers in total), 256 hidden size for input tokens, and 3 number of grouping iterations. The learnable token in MAP head has 512 dimensions. There are ∼10M parameters in this PGT-tiny.

**Group token layouts.** Given a fixed number of budget on group tokens, we explore three choices on how they should be arranged across grouping blocks and layers: descend, flat and ascend. Intuitively, more group tokens will have higher capacity of capturing smaller parts and detailed visual features, while less group tokens are more prone to carry global information. As shown in table 2 bottom row, flat or descend number of group tokens performs the best. In practice, we find that using flat (same number of group tokens in three grouping blocks) version achieves better training stability.

**Group token dimension shapes.** Similar to token number arrangements, we explore how group token dimensions should be set. Under three choices, progressively increasing the dimension size in the later layers performs the best, shown in first row of table 2. This also aligns with the intuition that later layers contain more information and requires higher capacity to represent groups.

**Multi-grouping vs single grouping.** We further test whether multi-head grouping helps improve performance. As a fair comparison, we use 6 heads and 128 group tokens per head for a multi-grouping model, and 1 head with 6×128 group tokens for a single grouping model. We find that adopting multi-head design can improve the performance from 62.2% to 66.3%, a 4.1% accuracy boosts, showing that having multiple heads indeed helps with representation learning.

**Grouping distribution entropy.** Will grouping process collapse to some specific group token during training? We visualize the entropy of marginal distribution over tokens $p(c)$ and conditional distribution $p(c|x)$ in figure 4. Interestingly, we observe that conditional probability, i.e. the assignment to group tokens, tends to become more certain during training, while the marginal distribution remains having descend entropy, indicating collapses not happening in training.

**Peak memory usage.** As discusssed in section 3.4, given the same number of tokens, the grouping operation uses less memory than the self-attention operation. We show the percentage of peak memory usage in PGT$_G$-B compared to ViT-B with the same patch size (4×4) in table 3. The

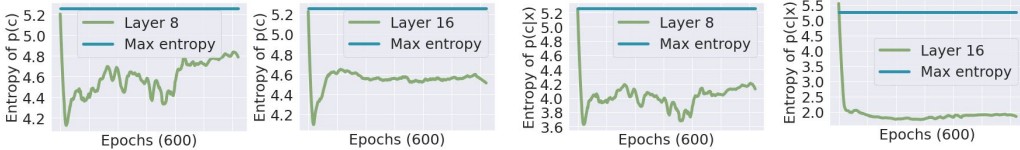

Figure 4: The entropy curves of grouping distributions $p(\boldsymbol{c})$ and $p(\boldsymbol{c}|\boldsymbol{x})$ across different layers.

usage is obtained from the forward inference graph, as in practice the underlying complex hardware optimizer is a less accurate measurement and varies across infrastructures.

| #group tokens | 16 | 32 | 64 | 128 | 256 | 384 | 512 | 768 | 1024 | ViT-B |
|---|---|---|---|---|---|---|---|---|---|---|
| Peak memory(%) | 4.6 | 4.6 | 4.6 | 4.6 | 4.6 | 6.1 | 8.2 | 12.2 | 16.3 | 100 |

Table 3: Peak memory usage of PGT-B compared to the baseline model ViT-B with $4 \times 4$ patch size.

### 4.3 OUT-OF-DISTRIBUTION ADAPTIVE COMPUTATION

One surprising and powerful ability of PGT is adaptive computation. For example, given a model trained using $M_1$ group tokens per layer, one can choose to use $M_2$ group tokens in inference, where $M_2 \neq M_1$. This is because that the initial seeding group tokens are drawn from a probabilistic distribution, and the number of samples can be customized. This property leads to a highly customizable inference without re-training the model. When $M_1 \neq M_2$, the model copes with an out-of-distribution (OOD) problem where test time setting is different from training. We observe surprisingly strong generalization with our model. *Specifically, with more tokens $M_2 > M_1$ in inference, the performance can actually outperform the setting ($M_2 = M_1$) used in training, even if it is OOD for the model.*

The results for OOD adaptive computation are summarized in table 4. We mainly test PGT$_G$-Tiny with a grid evaluation that varies the number of group tokens in training $M$ and the number of group tokens in inference $N$, and also show the main model's results in the last row. When using the main model PGT$_G$-B to perform adaptive inference, with only 12.5% of the number of group tokens compared to training, the performance can still be maintained at 72.1% with only a $\sim$8% drop on top-1 accuracy. The adaptive computation ability is important for both general image understanding where images have varying number of objects and need different numbers of groups, and scenarios where test-time computational resource is constrained. This flexibility is an important advantage that grouping backbones hold.

| tr/inf | 16 | 32 | 64 | 128 | 256 | 384 |
|---|---|---|---|---|---|---|
| PGT$_G$-Ti-16 | $\underline{57.4}$ ($\times 1$) | 58.3 ($\times 2$) | **58.5** ($\times 4$) | 58.5 ($\times 8$) | 58.5 ($\times 16$) | 58.4 ($\times 24$) |
| PGT$_G$-Ti-32 | 57.3 ($\times \frac{1}{2}$) | $\underline{59.9}$ ($\times 1$) | 60.8 ($\times 2$) | **61.0** ($\times 4$) | 61.0 ($\times 8$) | 60.9 ($\times 12$) |
| PGT$_G$-Ti-64 | 53.0 ($\times \frac{1}{4}$) | 59.2 ($\times \frac{1}{2}$) | $\underline{61.7}$ ($\times 1$) | 62.6 ($\times 2$) | 62.9 ($\times 4$) | **62.9** ($\times 6$) |
| PGT$_G$-Ti-128 | 44.9 ($\times \frac{1}{8}$) | 56.6 ($\times \frac{1}{4}$) | 61.8 ($\times \frac{1}{2}$) | $\underline{63.9}$ ($\times 1$) | 64.7 ($\times 2$) | **64.8** ($\times 3$) |
| PGT$_G$-Ti-256 | 27.2 ($\times \frac{1}{16}$) | 47.4 ($\times \frac{1}{8}$) | 58.8 ($\times \frac{1}{4}$) | 63.3 ($\times \frac{1}{2}$) | $\underline{65.1}$ ($\times 1$) | **65.5** ($\times \frac{3}{2}$) |
| PGT$_G$-Ti-384 | 26.1 ($\times \frac{1}{24}$) | 43.0 ($\times \frac{1}{12}$) | 55.4 ($\times \frac{1}{6}$) | 61.7 ($\times \frac{1}{3}$) | 64.6 ($\times \frac{2}{3}$) | $\underline{65.5}$ ($\times 1$) |
| PGT$_G$-B-256 | 60.4 ($\times \frac{1}{16}$) | 72.1 ($\times \frac{1}{8}$) | 77.1 ($\times \frac{1}{4}$) | 78.9 ($\times \frac{1}{2}$) | $\underline{79.7}$ ($\times 1$) | **79.9** ($\times \frac{3}{2}$) |

Table 4: Out-of-distribution adaptive computation by selecting different numbers of initially sampled tokens. Row: number of tokens used for training. Column: number of tokens used for inference. Top-1 accuracy is reported under linear evaluation protocol using ImageNet-1K. The reported performance of first six rows is obtained using a tiny version of PGT, and last row is the main model. Number of group tokens is the same for underlined numbers in training and inference. **Bold numbers** are the best results.

### 4.4 DOWNSTREAM TASK TRANSFER: SEMANTIC SEGMENTATION ON ADE20K

To evaluate the generalizability of pretrained feature produced by PGT, we test the transfer performance of semantic segmentation with ADE20k. Following the standard setup, we finetune our

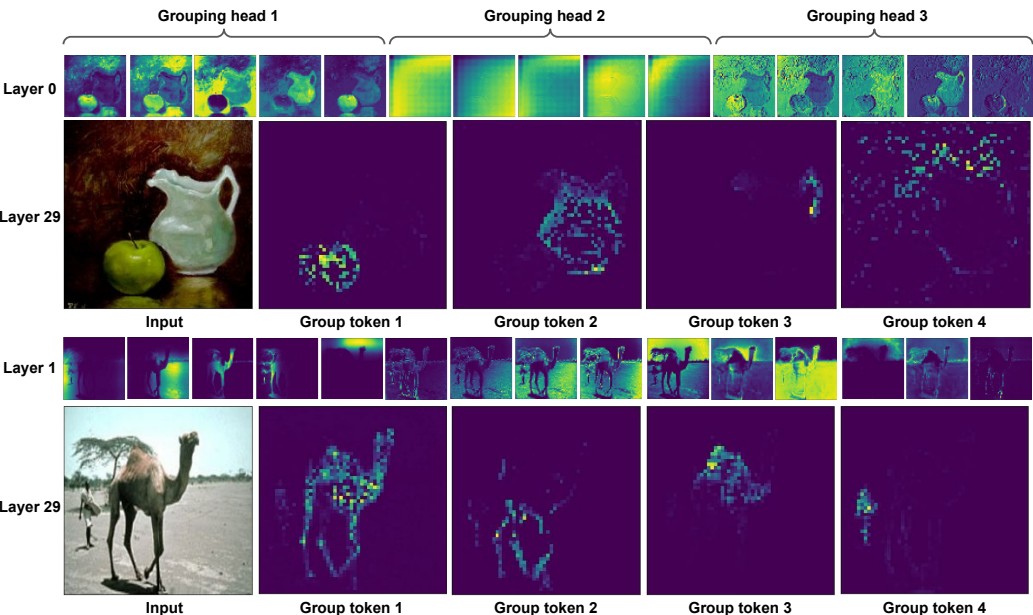

Figure 5: Visualization of attention maps of each group tokens across layers and grouping head. $L$ indicates layer indices. Five group tokens for each grouping head. Smaller images are for early layers, arranged as five group tokens per grouping head. Large images are for the last layer.

model with the same data augmentation for 128 epoch. The baseline method uses DINO + ViT-B/16 (Zheng et al., 2021). For our model, we add one linear classification layer after the pre-trained $PGT_G$-B for fine-tuning. To adapt to more objects and complex scenes in the segmentation datasets, we use 1024 group tokens for inference, benefiting from the adaptive computation ability of our model. We find that our model can obtain 45.1% on mean IoU while the baseline achieves 44.1% (Bao et al., 2021), leading to a 1.0% improvements.

### 4.5 Grouping Visualization

We visualize the attention maps calculated between group tokens and input tokens in figure 4.5. We find that (1) using multiple grouping heads can capture different information within each head. For example, in layer 0, the first head captures light and color, second head focuses on only spatial locations, and the third head potentially relies on textures; (2) group tokens can capture different semantic parts, for example, in the first image, group tokens separate apple, jar, handle, and background. In the second image, camel, legs, camel hump, and human are separately grouped. Compared to standard ViT in DINO (Caron et al., 2021) where only a single foreground can be extracted using [CLS] token, our model can flexibly group different parts given an image, leading to a set of tokens that are potentially more meaningful and customizable. Note that the grouping results are still different from human's vision, and sometimes generates parts that seem to be "fragmented". This is possibly due to the "parts-to-whole with data augmentation" training loss. Human vision, in contrast, is sensitive to moving objects and trained within a 4D space. Nevertheless, we believe with a similar dataset, environment and loss design, our grouping model can potentially produce groupings more coherent and sensitive to boundaries and moving objects.

## 5 Conclusion

In this paper, we propose Perceptual Group Tokenizer (PGT), a new visual recognition architecture entirely built through perceptual grouping principles. The proposed model shows strong performance on self-supervised learning benchmark ImageNet-1K with linear probe evaluation, and has desirable properties such as adaptive computation and high model interpretability in each operation. This work can enable a new paradigm for designing visual recognition backbones, and we hope to inspire more research progress along this direction. One limitation of the proposed model is its relatively expensive computation cost due to the iterative grouping processes. This can be potentially addressed by other grouping operations, such as those grouping operations with closed-form solutions, which is a promising direction for the future work.

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

# A APPENDIX

## A.1 LEARNABLE SAMPLING DISTRIBUTIONS

Our proposed Perceptual Group Tokenizer (PGT) model initializes a set of group tokens through sampling from a distribution. This set of group tokens then serve as the initial "seeding" for the grouping process. We explore two methods to serve as the initial distribution: learnable Gaussian distribution and Normalizing Flows. We would like the extra cost of the grouping process to be minimal, therefore, use two light-weighted versions.

### A.1.1 GAUSSIAN

Similar to the standard usage of learnable Gaussians in generative model literature (Kingma & Welling, 2022; Ho et al., 2020), we use the reparameterization to perform a learnable sampling process: $c = \mu + \sigma * \epsilon$, where $\epsilon$ is drawn from a unit Gaussian $\mathcal{N}(0, I)$

### A.1.2 FLOW

As Gaussian distribution might have limitations in covering complex distribution shapes, especially in the high-dimentional space, we also explore a version with one step of affine coupling flow transformation (Dinh et al., 2016). Since we only require the differentiable sampling procedure and do not need to compute the determinant of Jacobian matrix, we directly apply the transformation without splitting the dimensions by half:

$$c = a * \epsilon + b \tag{6}$$
$$(\log s, t) = \mathrm{MLP}(c) \tag{7}$$
$$s = \exp(\log s) \tag{8}$$
$$c = s * c + t \tag{9}$$

where $\epsilon$ is drawn from a unit Gaussian $\mathcal{N}(0, I)$. This transformation is simply a re-scaling and translation (similar to Gaussian) but conditioned on per sample $\epsilon$. More details are in (Dinh et al., 2016; Kingma & Dhariwal, 2018). We only apply one step of this transformation, leading to minimal parameter increase and negligible inference time difference.

## A.2 MODEL ANALYSIS

In this section, we add more analysis on our model's performance and computational costs.

### A.2.1 GROUPING ENTROPY

**Grouping distribution entropy.** The main paper has discussed and shown the grouping distribution entropy curves on several layers. In the appendix, we demonstrate curves from more layers in figure 6 and figure 7, where the first one is marginal distribution and second one is conditional distribution.

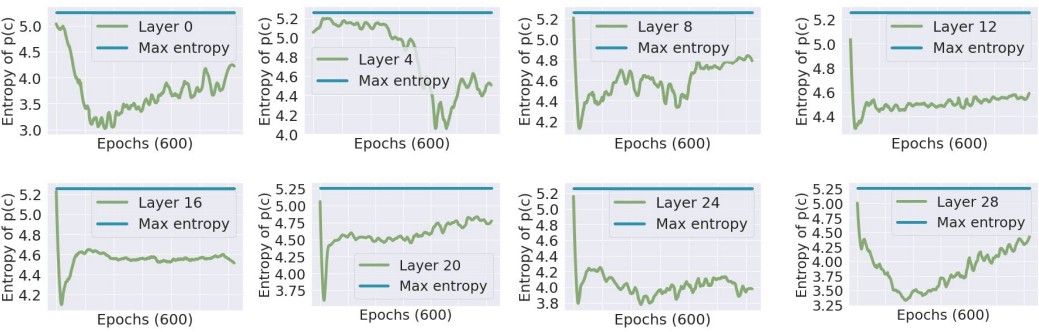

Figure 6: The entropy curves of marginal distribution $p(c)$ grouping across different layers.

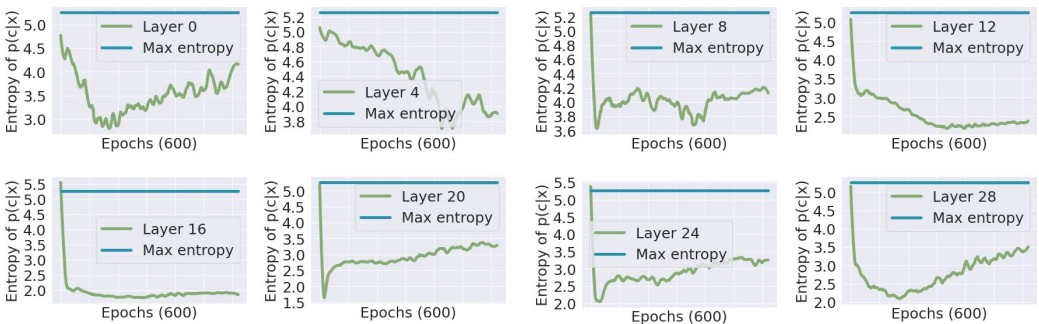

Figure 7: The entropy curves of conditional distribution $p(c|x)$ grouping across different layers.

### A.2.2 GROUPING ITERATIONS

In our backbone, we find that more grouping iterations will lead to a better performance. We explore the number of grouping iterations on the PGT-Tiny model and PGT$_G$-B-256. On the tiny version, we find the model achieves 61.4, 63.8, and 65.1 on the linear probe evaluation with number of interations is 1, 2, and 3. For the main model, the performances are 79.3, 79.6, and 79.7 respectively. The model's increased depth potentially helps with the lack of grouping iterations in the deep model. But in general, having the grouping process is still important in obtaining higher performance.

### A.2.3 INFERENCE TIME

We also profile our model's inference time, compared with ViT-B with 4x4 patches (the same amount tokens) for ablation study on the grouping operation. Note that our model and framework are built upon a complex infrastructure that uses XLA and other hardware accelerator to optimize speed. We find varying number of group tokens only lead to small influence. PGT-B-256 has  640 im/sec/core and ViT-B/4 has  680 im/sec/core. Using smaller number of grouping iterations can speed up the inference to  710 im/sec/core (2 iter2) and  820 im/sec/core (1 iter).

Note that this is only due to the specialty of the underlying infrastructure. In general, having less number of group tokens should still increase the inference speed, since the attention operation is a key computation bottleneck for vision models.

### A.2.4 GFLOPS

In table A.2.4, we show the gflops for our model under various inference budgets. Note that, as pointed in other works (Dao et al., 2022), gflops often do not fully reflect the model's computation performance. Due to that our model needs iterative grouping process, it'll increase the gflops count. But as shown in peak memory usage and inference time, the model's computation costs are either similar are much less.

| #group tokens | 16 | 32 | 64 | 128 | 256 | 384 | 512 | 768 | 1024 | ViT-B |
|---|---|---|---|---|---|---|---|---|---|---|
| Gflops | 99.2 | 131.1 | 194.9 | 322.6 | 577.8 | 833.1 | 1088.3 | 1599.0 | 2109.3 | 451.6 |

Table 5: Gflops of PGT-B compared to the baseline model ViT-B with the same patch size.

### A.2.5 PROBABILISTIC PERSPECTIVE OF GROUPING OPERATIONS

Due to the probabilistic nature, our model is also quite compatible with a full "treatment" with the variational inference framework, which can provide certain backup for our grouping operations already in the current model. We can treat the group token embeddings $c$ as the latent variables, where the grouping process uses iterative amortized inference (Marino et al., 2018) to refine the latent variable. The grouping modules, including GRU, MLP, attention, and other layers are designed

to better infer the embeddings (latent variables). The training signal is a pragmatic loss (instead of reconstruction loss), which has been demonstrated in (Reddy et al., 2021; Alemi et al., 2016). The key differences are: (1) there is no sampling in each inference step; (2) the regularization from unit Gaussian distribution is set to zero. We do believe a full probabilistic treatment of the perceptual grouping architecture can be a very interesting next step.

### A.2.6 MORE VISUALIZATIONS

In this section, we show more visualizations of the attention maps for generated group tokens by Perceptual Group Tokenizers in figure 8, 9 and 10.

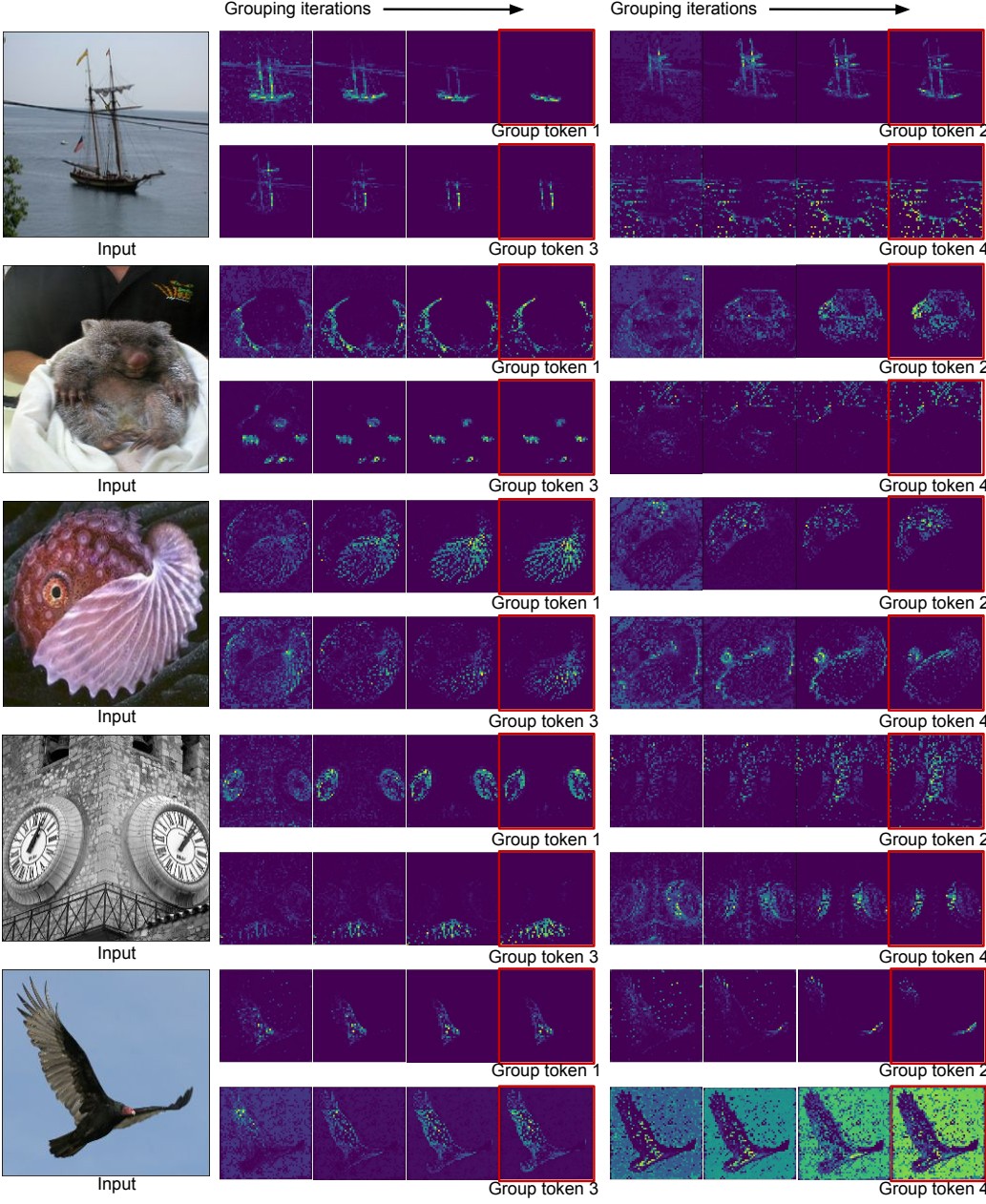

Figure 8: Visualization for attention maps of group token samples during the grouping process. PGT uses 256 group tokens in inference time.

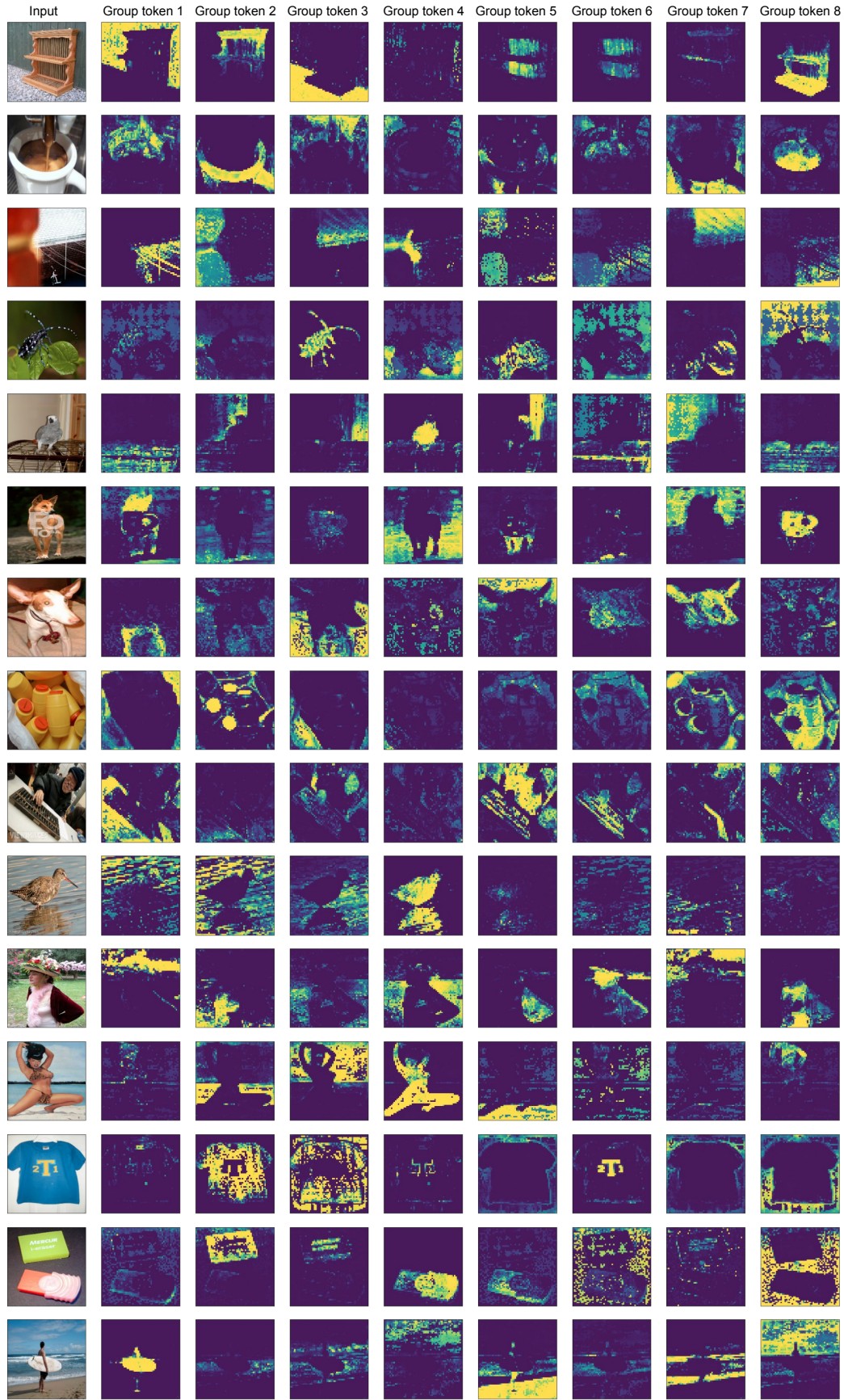

Figure 9: Grouping results from the 21st layer, using 8 group tokens in inference time.

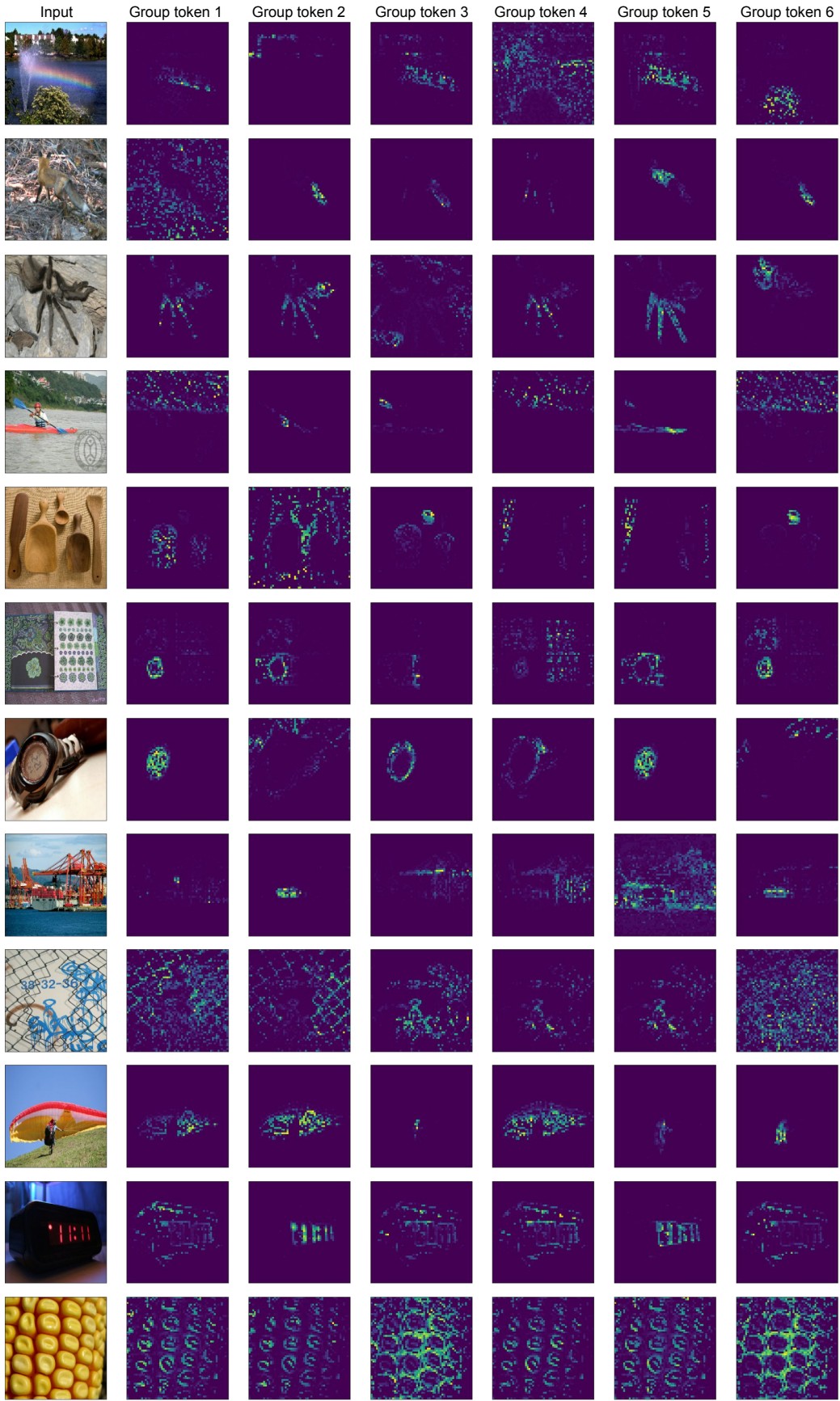

Figure 10: Grouping results from the last layer, using 256 group tokens in inference time.

