# OpenReview forum: "Perceptual Group Tokenizer: Building Perception with Iterative Grouping"
_ICLR.cc/2024/Conference — ICLR 2024 poster_

### Official Review · Reviewer_ABtJ · 2023-10-28

**Soundness:** 2 fair
**Presentation:** 3 good
**Contribution:** 3 good
**Rating:** 5
**Confidence:** 5

**Summary:**

This paper proposes the Perceptual Group Tokenizer model (PGT), which utilizes grouping operations for visual feature extraction. It demonstrates competitive performance in self-supervised learning while reduces the computation complexity into O(n*m) compared with the complexity O(n^2) of vision transformer. The PGT also offers adaptive computation without re-training via flexible number of grouping tokens, and offers interpretability in feature representations.  Quantitative experiments and visualization demonstrate the PGT's effectiveness.

**Strengths:**

1. The idea of Perceptual Group Tokenized is intrersting. By alternatively refining the group tokens and image feature tokens, the computation complexity has been reduced comparing with self-attention.
2. The adaptive computation ability of PGT is good, which could be used to meet the needs of different inference speeds.
3. The interpretability of model is relatively good.

**Weaknesses:**

1.  The experiment is not sufficiently comprehensive and the results are weak. Since a new model architecture is proposed, the performance under both supervised learning and self-supervised learning should be presented to show its’ universal. In performance comparison, the listed comparable architectures should be compared with the same pre-training/supervised-training strategy, a similar amount of parameters. It will be better to do experiment with more model sizes(e.g, PGT-B, PGT-S,PGT-Ti or more ) and compare with counterparts under different sizes. The performance on downstream tasks are not clear, although segmentation performance is reported, more comparisons with relative methods are not sufficient.
2.  Besides the number of parameters, for fair comparison, the computational cost(Gflops or MACs) should be shown. The inference time is missing, which is important for evaluating models’ efficiency.

**Questions:**

1. Do the group tokens in each block generate independently? If not, maybe the relavant description is ambiguous. If so, why don't the group tokens take its' output of the previous block as the next block's input?
2.  Could the iterative grouping processes be considered multiple cross-attention between input tokens and group tokens? What's the difference or advantages of the iterative grouping processes compared with cross-attention? How much does the number of grouping iterations matter?

---

> ### Author Response · Authors · 2023-11-23
> **Thank you for the feedback**
>
> We appreciate the reviewer’s feedback and comments. Specific questions are addressed below.
>
> **W1.1. Multiple model sizes**
>
> We agree and are working on the results with various model sizes, the performance of variants will be included in the final version. Providing one data point here, we develop a PGT-S with 34M parameters, the model is still in early training (1/3 course) and show similar performance (76.8%) with PGT-B (77.3%)
>
> **W1.2 Results and others**
>
> We have updated our results in the paper. The table 1 is arranged for more clear fair comparison. Note that the performance of 80.3% is a strong result under the perceptual grouping paradigm, especially, we show that PGT has a much lower memory cost in inference.
>
> **W2. Profiling computational costs and stats**
>
> Following the reviewer’s suggestion, we have updated our paper with following modifications: peak memory usage in table 3, inference time in A.2.3, and Gflops in A.2.4. Especially to highlight, PGT has a very low memory cost compared to ViT under fair comparison, summarized in table 3.
>
>
> **Q1. Independent group tokens**
>
> This is a great question. Yes, each layer’s group tokens are generated independently. Our paper aims to show that grouping operations can be used as a more general operator to exchange information among input tokens, compared to self-attention. Both operations are per-layer refinement. Adding cross layer connections can complicate the process.
>
> Regardless, we indeed have tried reusing the previous layer’s group tokens as a variant (e.g., residual connections, or cross attention), and found that the performances are similar. This indicates that the grouping results are already stored in the refined patch token embedding vectors and can be easily recovered in the new round.
>
> **Q2. How to consider grouping**
>
> Grouping uses attention, but performs softmax in the query (group tokens) axis, different from cross attention (in the key axis). This ensures that each operation is an assignment. We have explored using key axis softmax and found the performance dropped a lot.
>
> Grouping iterations are very important in general. We explain some results in the appendix, A.2.2. Empirically, having a sufficient amount of binding process (e.g. 3) will always help on the performance, especially when the model depth is not too high. We’ve also tried using learnable tokens each layer to perform one-step cross attention and found it leads to strong overfitting.
>
> Instead of viewing grouping as a deterministic procedure (like cross attention), we think it’s better to view it from a probabilistic perspective, where there is a distribution over group tokens that summarize the tokens. With a better distribution, the performance can be boosted. For example, we add one more experiment where the initial Gaussian sampling distribution is replaced by a more advanced Normalizing Flow sampler. The normalizing flow models a more flexible initial distribution that better covers the space. We found that it can lead to an increase in the final performance.
>
> We appreciate your comments and hope the above answers address the reviewer’s questions and look forward to your response. Please let us know if you have more questions!

---

### Official Review · Reviewer_TtGg · 2023-10-29

**Soundness:** 3 good
**Presentation:** 3 good
**Contribution:** 3 good
**Rating:** 8
**Confidence:** 4

**Summary:**

The authors propose a self-supervised learning method that relies only on grouping operations. They call their model the Perceptual Group Tokenizer (PGT). The model's linear probe performance is on par with other state-of-the-art models such as ViT. It also produces a highly interpretable representation and shows some interesting properties like being somewhat adaptable at inference time. Overall, the PGT demonstrates that grouping operations alone can produce a rich visual representation.

**Strengths:**

The writing of this work is very clear. The authors do a great job of motivating the problem and contextualizing its significance. As the authors note, perceptual grouping has historically been an important concept in computer vision, but it had not been proven to be as powerful as feature detection. This work proposes a novel way to use grouping principles for self-supervised representation learning on large-scale natural image data.

The approach also appears to be well done. The analyses of number and size of grouping are also insightful, and demonstrate a thorough evaluation of their model. Making the connection to self-attention in ViT is also an important and original contribution for understanding why both the PGT and ViT work.

**Weaknesses:**

One of the main weaknesses I see is that the PGT performs very similarly to the ViT. As the authors note, their method is in some sense a more general version of the self-attention approach so it is somewhat unsurprising that performance is on par with ViT. I think the paper could be strengthened by more discussion about why the perpetual grouping tokenizer might be better or more useful than other methods like ViT.

Along similar lines, I think the interpretability results in Figure 6 and section 4.5 could be better contextualized. It is not immediately obvious how the interpretability conveyed by the attention maps compares to other state-of-the-art models. In addition, this analysis could be strengthened by a discussion of how the attention maps at each stage do or do not agree with what we would expect of human vision.

**Questions:**

As I mention in the weaknesses, could the authors touch on what are the benefits of the PGT over ViT given that they perform similarly on the ImageNet 1k linear probe?

In a similar vein, I would also like some of the model interpretability results to be expanded on in the context of other state-of-the-art methods as well as human visual perception.

Finally, I thought adaptive computation (Section 4.3) was a bit too brief for me to appreciate. Could the authors explain what they mean by adaptive computation and why it is a benefit of PGT?

Generally, I enjoyed this paper. I am open to increasing my score if these concerns are addressed,

---

> ### Author Response · Authors · 2023-11-23
> **Thank you for the constructive feedback!**
>
> We appreciate your insightful feedback and would like to thank the reviewer for your comments and acknowledgement on our work. We address the questions below.
>
> **W1&Q1. Compared to ViT**
>
> This is a great question. We answer in following perspectives:
>
> - ViT is a grouping with a fixed number of grouping slots by design (e.g. the patches), this limits its generalizability across other vision tasks, where larger resolution is needed, or many frames need to be processed (e.g. videos). PGT backbone separates the concept of “group tokens” out, making it much more capable of handling a broader class of vision tasks without the need to adapt the model to a more efficient version.
>
> - PGT has the potential to generate better tokens. ViT and ConvNet have shown similar results in classification, but ViT is more favored due to its meaningful token and easier alignment with language in multimodal training. PGT, as a more flexible model, can combine patches into regions and produce better tokens. This can be even more true in videos, where patches of objects can be grouped and tracked across frames.
>
> - PGT has a probabilistic nature in each grouping process, naturally adapts to different number of group tokens without re-training.
>
> **W2&Q2. Visualization interpretation.**
>
> Thanks for the suggestion! We have added text in the paper, and also explain here for easier access.
>
> Highlighting a key difference compared to ViT, we find that the grouping process can automatically generate object groupings and segments corresponding to different group tokens, while a standard ViT in DINO can only extract a single foreground using [CLS] token without relying on other adaptations. Using multiple learnable embeddings recently show similar effects[a] with our model, but it can potentially be less flexible and adaptive with various number of group tokens due to a lack of probabilistic nature.
>
> Our model shows both human-like and fragmented-to-human grouping results. In the current SSL framework, the loss uses “parts-to-whole” or “parts-to-parts” matching and encourages alignment within the same 2D image. This either leads to object confusion when multiple objects appear, or a lack of sense of clear object boundaries since no temporal information is used. Human vision is sensitive to moving objects and trained with 4D space. We believe with a similar dataset, environment and optimization loss design, e.g. training vision models with time axis and interactions, our grouping model can potentially produce groupings more similar to human's.
>
> **Q3. Adaptive computation.**
>
> In the general image domain, where images are not curated, the number of objects in the same image can vary. PGT has the adaptive computation ability, where, during training, only a fixed number of group tokens (e.g. 256) is used, in inference time the trained model can use a different number of group tokens to produce image features and group proposals.
>
> This is because of the probabilistic nature of PGT: initial group tokens are sampled from a distribution, and one can sample as many tokens as possible given a distribution. If using a fixed number (e.g., K) of learnable embeddings, it cannot go above the fixed number (K). We also observe that this adaptivity and robustness can come from using sampling during training, which introduces random noises in the grouping procedure.
>
> [a]. Vision Transformer Needs Registers, Timothée Darcet, Maxime Oquab, Julien Mairal, Piotr Bojanowski
>
> We appreciate your suggestions and hope the above answers address the reviewer’s questions and look forward to your response. Please let us know if you have more questions!

---

> > ### Comment · Reviewer_TtGg · 2023-12-02
> > **response to author comments**
> >
> > Thank you for the thoughtful comments, and my apologies for not responding sooner. I feel that my questions have been adequately addressed. I am also not overly concerned about the "weak performance" noted by other reviewers. I think the work provides valuable insights into importance of grouping regardless of not achieving state of the art. I will raise my score to 8.

---

### Official Review · Reviewer_P2FR · 2023-10-31

**Soundness:** 3 good
**Presentation:** 3 good
**Contribution:** 2 fair
**Rating:** 6
**Confidence:** 4

**Summary:**

This paper proposes the Perceptual Group Tokenizer (PGT), a ViT-like architecture that clusters tokens to implement the principle of perceptual grouping for visual recognition. Specifically, PGT combines a slot attention-like token grouping module within ViT blocks.

**Strengths:**

- The paper is generally well-written.
- Prior work on feature detection and perceptual grouping is well-discussed, although recent works on token grouping are missing.

**Weaknesses:**

**Limited technical novelty**

The proposed method is essentially a combination of ViT and slot attention (or its token grouping variants).
The concept of token grouping has been extensively studied in prior work. Please refer to the related work section of ToMe [1] and CoCs [2] for examples.
On the other hand, DINOSAUR [3] combined a ResNet/ViT encoder with slot attention to scale up object-centric learning for real-world images, which is also relevant to this work.

[1] Token merging: Your vit but faster. ICLR'23.\
[2] Image as Set of Points. ICLR'23.\
[3] Bridging the Gap to Real-World Object-Centric Learning. ICLR'23.

---
**Unclear empirical benefits**

In addition to the technical novelty, the empirical merit of the proposed method is unclear.
1. Performance: It is not better than prior works, such as DINO.
2. Efficiency: Several efficient ViT-based works exist, such as ToMe, DynamicViT [4], A-ViT [5], etc.
3. Adaptive computation w/o retraining: ToMe claims to offer the same benefit.
4. Grouping visualization: Other grouping methods also offer similar advantages (Fig. 4 of ToMe). CAST [6] is even better in this regard.

[4] DynamicViT: Efficient Vision Transformers with Dynamic Token Sparsification. NeurIPS'21.\
[5] A-ViT: Adaptive Tokens for Efficient Vision Transformer. CVPR'22.\
[6] CAST: Concurrent Recognition and Segmentation with Adaptive Segment Tokens. arXiv'22.

**Questions:**

There are many similar works in recent years. What are the substantial differences or unique advantages of this work?

---

> ### Author Response · Authors · 2023-11-23
> **Thanks for the review**
>
> We appreciate the reviewer’s feedback and comments. A new revision is submitted with the updated draft. We address the specific questions below.
>
> **W1. Concerns about the novelty of the paper**
>
> We disagree with this point. The mentioned papers are relevant, but we are delivering fundamentally different messages in our paper, which we clarify below:
>
> - The core message
>
> We replace the self-attention operation with grouping operations, showing that grouping operations can perform feature learning in self-supervised learning. This alone separates our paper with other works [1,3,4,5,6], which still rely on ViT’s self attention operations or convolution layers.
>
> Specifically, [1,4,5] are ad-hoc adaptations for pruning ViTs and orthogonal to our work. The techniques like patch token drop can directly be applied to PGT input tokens x as an efficiency improvement method. CAST[6] uses object proposals from traditional vision methods as input, this makes it not comparable to most backbones, including PGT. [2] is indeed relevant, but focuses on supervised learning, uses non-standard operations (fixed center feature pooling, aggregation with sigmoid activations, patch pooling for reduction, etc.).  Instead, in our paper, we are the first to cleanly use *iterative grouping to replace the self attention* operation and show a strong performance.
>
> We also show that the grouping tokens as communication channels among input tokens is a more general form of information-exchange operator. This generalizes self-attention and can potentially lead to more advances in vision models.
>
> We do not think the mentioned works [1-6] deliver these messages.
>
> - Self-supervised learning
>
> Perceptual grouping is a general vision framework with a long history, and the root for many vision concepts and algorithms, *especially unsupervised*. There is a natural connection between pixel similarity measurement, grouping, and providing training signals from images themselves. Many classical vision tasks, such as segmentation, tracking, and 3D multiview problems can be formulated as grouping processes. Showing grouping works, especially under self-supervised learning framework, can be a critical stepstone to unify many vision tasks.
>
> - We do not combine ViT and Slot attention.
>
>  The main components of ViT are tokenization, self attention, and mlp, where self attention is the core. We use tokenization, grouping and mlp. There is no ViT in our model.
>
> Slot attention combines the grouping procedure with ConvNet, showing the feature maps from the last layer of convolutional nets can be grouped. We instead show grouping can be used as a core operation to drive the entire feature learning.
>
>
> We appreciate that the reviewer raises these relevant and inspiring works, and have added paragraphs discussing them in detail in section 2.
>
>
>
> **W2. Empirical benefits**
>
>   Note that we do not aim to propose an all-round model that beats every possible aspect of state-of-the-arts from all related works, and most papers do not either. As discussed above, the mentioned papers [1,4,5] are orthogonal to our work and can be applied to PGT. The intriguing part of PGT is that the performance, memory efficiency, adaptive computation and interpretability are all in a single model cleanly driven by one principle.
>
> We hope the explanation and response above address the reviewer’s questions.
>
> [1] Token merging: Your vit but faster. ICLR'23.
>
> [2] Image as Set of Points. ICLR'23.
>
> [3] Bridging the Gap to Real-World Object-Centric Learning. ICLR'23.
>
> [4] DynamicViT: Efficient Vision Transformers with Dynamic Token Sparsification. NeurIPS'21.
>
> [5] A-ViT: Adaptive Tokens for Efficient Vision Transformer. CVPR'22.
>
> [6] CAST: Concurrent Recognition and Segmentation with Adaptive Segment Tokens. arXiv'22.

---

> ### Comment · Reviewer_P2FR · 2023-12-04
> **Thank you for the rebuttal**
>
> I read other reviews and the rebuttal sincerely. As an architecture paper, I still feel this paper is yet another ViT alternative with advantages over others not clearly justified. However, after reading the discussion, I agree with the points regarding motivation, specifically, restoring the concept of perceptual grouping and suggesting an alternative to self-attention.
>
> I raised my rating to 6 because I agree that exploring new principles can provide valuable insights. However, future studies should justify the practical benefits of this new principle. Specifically, one of the strongest advantages of ViT is its scaling law [1,2], while many Transformer variants are known to fail at such scaling [3]. Investigating the scaling law of PGT should be a crucial future direction. If verifying such scaling is infeasible due to computational resources, PGT should demonstrate other advantages of perceptual grouping. Justifying these benefits would significantly strengthen the impact of this work, benefiting not only academics but also practitioners.
>
> [1] Scaling Vision Transformers. CVPR'22.\
> [2] Scaling Vision Transformers to 22 Billion Parameters. ICML'23.\
> [3] Scaling Laws vs Model Architectures: How does Inductive Bias Influence Scaling? arXiv'22.

---

### Official Review · Reviewer_qQDA · 2023-11-03

**Soundness:** 3 good
**Presentation:** 3 good
**Contribution:** 3 good
**Rating:** 6
**Confidence:** 3

**Summary:**

This paper proposes a perceptual group tokenizer, which just uses the grouping operations to extract visual features and perform self-supervised learning. The authors also explain the connection between the proposed perceptual group tokenizer and the self-attention. The experimental results show the performance is competitive with some state-of-the-art self-supervised methods.

**Strengths:**

1. The concept of perceptual group tokenizer is novel, and seems to enable the networks to have more good properties including interpretability and so on.

2. Discussion between the perceptual group tokenizer and the self-attention is interesting, and can provide a new vision for vision transformer design.

**Weaknesses:**

1. The motivation is not clear. Since we have powerful vision transformers already, what is the advantages of the proposed perceptual group tokenizer. The authors claim that the perceptual group tokenizer have good properties such as adaptive computation without re-training and interpretability. However, I don't see the experimental results or the visualization that can provides proof of this claim.

2. The performance is still concerned. Because we should also focus on the accuracy despite some good properties, the results shown in Table 1 demonstrate that the method cannot beat the baselines. Moreover, the state-of-the-art methods proposed in 2023 are not compared.

3. The explanations to the grouping operation should give more details. Since grouping seems to be a explicit operation, the implicit operations for grouping such as MLP should show the correlation of the term "grouping".

**Questions:**

See weakness.

---

> ### Author Response · Authors · 2023-11-23
> **Thank you for the review**
>
> We appreciate the reviewer’s feedback and comments. A new revision is submitted with the updated draft. We address the specific questions below.
>
> **Q1.1. Motivation**
>
> As mentioned in the introduction, our motivation and contribution is on the fundamental mechanisms of vision backbones. We would like to both refer the reviewer to our first half of introduction, and explain the motivation here.
> - Fundamental paradigms.
>
> Perceptual grouping in computer vision has deep roots and many vision concepts/methods are developed under that paradigm, for example, segmentation, object tracking, action recognition, etc. Showing that by simply applying one principle - perceptual grouping - can lead to state-of-the-art performance, is very encouraging. We believe this principle can be transferred to related fields and applications given their deep connections with grouping.
>
> - link to ViT
>
> As discussed in the paper, PGT with grouping can be considered as a more generalized form of ViT. ViT will always have fixed number of grouping slots (equal to number of patches), while PGT separates the concept out and allows for flexible customization on number of groups. This makes it more generalizable to a broader class of vision tasks which require flexible designs in number of objects/groups (e.g., tracking). We also build connections between grouping and self-attention (section 3.4), serving as a generalization in the operator design and an explanation on why ViT works. This perspective can spur more variations of grouping ops to go beyond self attention and advance the vision backbone designs.
>
> - Generating better tokens.
>
> One big advantage of ViT over ConvNet is its meaningful tokens[b], although it has been consistently shown that ConvNet can achieve similar performance[b]. The tokens can be easily aligned with language tokens in multimodal training[b]. Perceptual grouping models have the potential to produce even better tokens by design. The grouping process is constructed to produce a set of tokens that flexibly summarize the “detailed small patches” into tokens with more semantic meanings.
>
> - Efficiency.
>
> As shown in table 3, the peak memory usage of perceptual grouping models is much less compared to ViT. The memory advantage is critical in larger resolutions, videos or multiview images.
>
> **Q1.2 Adaptive computation and grouping results**
>
> Our results of adaptive computation are in section 4.3, specifically table 3.
>
> The grouping interpretability results are shown in figure 5 and 6. For example, in figure 6, each large attention image corresponds to a group token, an image is parsed into smaller meaningful regions, such as apple, jar and handle, or camel, camel legs and person. More explanation is in section 4.5 and figure 6 caption.
>
> **Q2. Performance**
>
> We have updated the results of our model. We better aligned with DINO’s preprocessing, where we previously omitted random blur. The current results have shown on par or better on top-1 accuracy.  Note that this performance is achieved with much less inference memory cost.
>
> We also would like to point out that only the lower half of table 1 are  fair comparisons. We fully follow DINO’s framework and only compare the backbone performance. The upper half are cited as reference for other backbone architectures. We cite works within the similar time range of DINO. With better loss functions and training procedure, PGT can potentially have better performance.
>
> **Q3. Grouping operation explanation**
>
> The grouping process can be considered as a deterministic approximation for variational inference framework, where the group token embeddings are the latent variables **Z**. The grouping modules, including GRU, MLP, attention, and other layers are designed to better infer the embeddings (latent variables). The training signal is a pragmatic loss (instead of reconstruction loss), which has been demonstrated in [c,d]. The key differences are: (1) there is no sampling in each inference step; (2) the regularization from unit Gaussian distribution is set to zero. We do believe a full probabilistic treatment of the perceptual grouping architecture can be a very interesting next step. An explanation is added in the appendix in section A.2.5.
>
> [a]. Self-attention in Vision Transformers Performs Perceptual Grouping, Not Attention. Paria Mehrani, John K. Tsotsos
>
> [b]. ConvNets Match Vision Transformers at Scale. Samuel L. Smith, Andrew Brock, Leonard Berrada, Soham De
>
> [c] Pragmatic Image Compression for Human-in-the-Loop Decision-Making. Siddharth Reddy, Anca D. Dragan, Sergey Levine
>
> [d] Deep Variational Information Bottleneck. Alexander A. Alemi, Ian Fischer, Joshua V. Dillon, Kevin Murphy
>
> We hope our answers address the reviewer’s questions. Looking forward to your response and please let us know if you have more questions!

---

> > ### Comment · Reviewer_qQDA · 2023-12-04
> >
> > Thanks for your response. I think my concerns are well addressed, so I increased my score to 6.

---

### Official Review · Reviewer_8m3b · 2023-11-07

**Soundness:** 4 excellent
**Presentation:** 3 good
**Contribution:** 3 good
**Rating:** 8
**Confidence:** 4

**Summary:**

This paper introduces the Perceptual Group Tokenizer (PGT), a novel vision model that relies on iterative grouping to extract visual features and learn representations in a self-supervised manner. PGT proposes to use grouping operations instead of self-attention layers in ViT, which yields a self-attention-free visual backbone. It demonstrates competitive performance on the ImageNet-1K benchmark. The model also shows properties like adaptive computation and high interpretability. The paper provides comprehensive analysis, ablation studies, and visualizations that underscore the model's capability and potential as a new paradigm in visual backbone architecture design.

**Strengths:**

1. I like the idea of grouping operations only, without self-attention. Using perceptual grouping is innovative and theoretically sound, offering a fresh perspective on representation learning and architecture design.
2. The adaptability of computation in inference mode is also interesting. Without re-training, the model could inference with different number of group tokens. Table 1 also shows the accuracy will increase as there are more group tokens
3. The visualization of the attention map is very interesting. It not only shows more iterations yield clearer grouping, but also shows different grouping heads kind of learning disjoint visual representations.

**Weaknesses:**

1. Only ViT-B level(70-80M parameter) model is reported. It would justify the effectiveness if the proposed architecture works when scaling the model size up.
2. The computation cost, peak memory usage, and inference speed comparison with ViT-B are not reported. It would be informative for readers how fast the PGT is since PGT doesn't have memory/computation demanding self-attention operations.

**Questions:**

1. The model uses patch size 4x4, which means the visual backbone only downsamples the image by 4. Intuitively, this model should be good at dense prediction tasks that require high feature resolution, e.g. semantic segmentation and object detection. The authors reported the results of semantic segmentation on ADE20K in Section 4.4, but it only outperforms the ViT-B by a smaller margin. I understand that the segmentation architecture is different. So it would be interesting to compare ViT-B vs PGT-B with the same segmentation architecture (linear classification layer).
2. Since PGT is self-attention-free, the computation cost is not quadratically increasing with the input resolution. But it is still reasonable to compare to ViT under the same resolution, for example, 16x16 patch size and 8x8 patch size. I am wondering whether authors have done this ablation.
3. As mentioned in the weakness section, it would be interesting to have the computation cost of PGT. As Table 1 shows, as we increase the number of group token from 256 to 768, the linear probe accuracy increases from 79.3 to 79.7 How about less than 256 tokens and more than 768 tokens? It would be insightful to have a graph of number of tokens vs accuracy and inference speed.
4. In Mask Autoencoder paper, researchers find that higher linear probing accuracy may be not necessarily stand for better representation. It would be also interesting to compare with MAE-ViT against PGT under the fine-tuning setting.

---

> ### Author Response · Authors · 2023-11-23
> **Thanks for the constructive feedback!**
>
> We would like to thank the reviewer for the detailed and valuable comments. A new revision is submitted with the updated draft. We address the specific questions below.
>
>  **W1. Scale up model sizes**
>
> We agree. We are working on more experiments to scale up the model to 300M or larger level sizes and will add the results to the final version (the training takes some time).
>
> **W2. Profiling PGT’s computational stats & Q3**
>
> Thank you for this suggestion! We have updated the draft and added peak memory usage in table 3 and inference time in appendix section A.2.3. For easier read, the results are also listed here:
>
> - Peak memory usage
>
> We show the percentage of peak memory usage in PGT compared to ViT-B with the same patch size (4×4).  Even with a large number of group tokens, due to less number of heads and total attention tokens, we still only use a much smaller percentage of memory compared to the ViT-B backbone. Note that the usage is obtained using forward inference graph, as in practice the underlying complex hardware optimizer is a less accurate measurement and largely varies across infrastructures.
>
>
> | # group tokens        | 16  | 32  | 64  | 128 | 256 | 384 | 512 | 768  | 1024 | ViT-B |
> |-----------------------|-----|-----|-----|-----|-----|-----|-----|------|------|-------|
> | Peak memory usage (%) | 4.6 | 4.6 | 4.6 | 4.6 | 4.6 | 6.1 | 8.2 | 12.2 | 16.3 | 100   |
>
> - Inference time
>
> Similarly, we profile our model’s inference time against ViT-B/4. Our model is built upon a complex infrastructure with XLA and other hardware optimizers. We find that ViT has ~680 im/sec/core inference speed, while PGT has ~640 im/sec/core. Note that PGT has a heavier process (3 steps of iterative grouping), and can still maintain similar speed with ViT. With fewer grouping iterations (1 and 2), our model can achieve ~710 im/sec/core and ~820 im/sec/core.
>
> Due to the underlying hardware speed optimizers, changing the number of group tokens currently leads to negligible influence. Theoretically, with less number of group tokens, the inference time should still be reduced. We will look into how to profile the model in a more toy and simpler environment.
>
> **Q1. Segmentation**
>
> We have an updated result of 45.1% for PGT. Using ViT with a linear layer, we got similar results (44.2). We completely agree on that having more dense patches should indeed lead to higher performance. One very likely reason is DINO’s loss is a whole-image holistic loss, where parts of the same image are always matched with each other. The images in ImageNet, although curated, still contain multiple objects in most of them. This loss design might limit the power of our grouping backbone. We think a dense loss function that uses “pixel-group assignment” will further unleash the performance.
>
> **Q2. 8x8, 16x16**
>
> Yes, we have been keeping track of model performance in different resolutions. Since one core of the grouping process is the ability of automatically grouping patches and generating more meaningful “regions” (compared to large strided patchification), we mainly test the model on small patches to allow bigger flexibility in the binding procedure. A simple attempt on 8x8 and 16x16 lead to 79.2 and 77.0. PGT-B uses 384 as patch embedding vector dimension and 384 as the hidden dimension for MLP (ViT uses 768 and 3072). Scaling it to the same dimension lead to a 0.4 improvement on each model.
>
> **Q4. Fine-tuning**
>
> This is a great question. We fine-tuned our PGT and achieved 82.3% (DINO has 82.8%). Interestingly, we find that the standard data augmentation trick mixup negatively affects the model’s performance. Mixup combines two images in a linear combination way and will blur the images. This seems to interfere the grouping process. PGT only uses randAugment[a] for finetuning. We believe perceptual grouping models have their own augmentation pipeline for achieving the best performance.
>
> [a] RandAugment: Practical automated data augmentation with a reduced search space. Ekin D. Cubuk, Barret Zoph, Jonathon Shlens, Quoc V. Le
>
> We appreciate your suggestions and hope the above answers address the reviewer’s questions and look forward to your response. Please let us know if you have more questions!

---

### Author Response · Authors · 2023-11-23
**Thanks to all reviewers for the constructive feedback**

Dear reviewers,

We would like to express our gratitude to the reviewers for their constructive comments. We have updated the revision with following changes:

- Expanded related work in section 2 (Reviewer P2FR)

- Updated experimental results in section 4.1, 4.3(Reviewer qQDA, Reviewer P2FR, Reviewer ABtJ)

- Added discussion on adaptive computation and grouping visualization in section 4.3, 4.5 (​​Reviewer TtGg)

- Added model profiling in section 4.2 and appendix (Reviewer 8m3b, Reviewer ABtJ)

- Discussed the interpretation of grouping in the appendix (Reviewer qQDA)

Please let us know if you have more questions.

---

### Meta-Review · Area_Chair_HttF · 2023-12-10

**Metareview:**

This paper proposes to bring perceptual organization into neural visual recognition by replacing ViT's self-attention with a grouping operation similar to the slot attention mechanism.

The strengths of the paper are on-par performance in self-supervised learning with reduced computational complexity, adaptive and flexible numbers of grouping tokens without re-training, and interpretability in terms of context clustering.

The weaknesses of the paper are questionable empirical benefits, under-performing the state-of-the-art on individual aspects of accuracy, efficiency, adaptive computation, and grouping.  It is also unclear whether demonstrated performance gains would still hold when the model scales up.

This paper has received 5 detailed reviews, to which the authors have provided further clarification and additional results, successfully raising final ratings to 8/8/6/5/6, based on which the AC recommends acceptance.

**Justification For Why Not Higher Score:**

Incremental technical novelty and relatively weak experimental validation.

**Justification For Why Not Lower Score:**

The paper is well motivated with a call for integrating perceptual organization into visual learning, which all the reviewers appreciate more over weak experimental performance.

---

### Decision · Program_Chairs · 2024-01-16

Accept (poster)